# Stigma experiences, effects and coping among individuals affected by Buruli ulcer and yaws in Ghana

Ruth Dede Tuwor[1,2], Tara B. Mtuy[3], Yaw Ampem Amoako[2,4]*, Lucy Owusu[2], Michael Ntiamoah Oppong[2], Abigail Agbanyo[2], Bernadette Agbavor[2], Michael Marks[3], Stephen L. Walker[3], Dorothy Yeboah-Manu[5], Rachel L. Pullan[3], Jonathan Mensah Dapaah[1], Richard Odame Phillips[2,4], Jennifer Palmer[3], for the SHARP collaboration[¶]

1 Department of Sociology and Social Work, Kwame Nkrumah University of Science and Technology, Kumasi, Ghana, 2 Kumasi Centre for Collaborative Research, Kwame Nkrumah University of Science and Technology, Kumasi, Ghana, 3 London School of Hygiene and Tropical Medicine, London, United Kingdom, 4 School of Medicine and Dentistry, Kwame Nkrumah University of Science and Technology, Kumasi, Ghana, 5 Noguchi Memorial Institute for Medical Research, University of Ghana, Accra, Ghana

¶ Membership of SHARP is provided in the Acknowledgments
* yamoako2002@yahoo.co.uk

**Data Availability Statement:** All relevant data are within the manuscript and its supporting information files.

## Abstract

### Background

Stigma related to skin neglected tropical diseases like Buruli ulcer (BU) and yaws has remained underexplored and existing studies are limited to individual diseases despite the WHO call for integration in disease management. Within two districts in central Ghana, we explored stigma associated with BU and yaws to understand overlaps and disease-specific nuances to help guide integrated interventions.

### Methodology/Principal findings

In-depth interviews were conducted with 31 current or formerly affected individuals to assess the experiences, effects and coping strategies adopted to manage disease related stigma. Data were analysed along broad themes based on the sociological construct of macro and micro interaction and Goffman's treatise on stigma.

Disapproving community labels fueled by misconceptions were noted among BU participants which contributed to macro stigma experiences, including exclusion, discrimination and avoidance. In contrast, a high level of social acceptance was reported among yaws participants although some micro-level stigma (anticipated, felt and self-stigma) were noted by individuals with both diseases. While younger participants experienced name-calling and use of derogatory words to address affected body parts, older participants and caregivers discussed the pain of public staring. Stigma experiences had negative consequences on psychosocial well-being, schooling, and social relations, particularly for BU affected people. Problem-focused strategies including confrontation, selective disclosure and concealment as well as emotion-focused strategies (religious coping and self-isolation) were noted.

**Funding:** This project is supported by the National Institute for Health and Care Research (NIHR) https://www.nihr.ac.uk/ under its Research and Innovation for Global Health Transformation (RIGHT) Programme [Grant Reference Number NIHR200125 to ROP, MM, DYM, RLP and SW]. The funders had no role in study design, data collection and analysis, decision to publish, or preparation of the manuscript.

**Competing interests:** The authors have declared that no competing interests exist.

## Conclusions and significance

The types and levels of stigma varied for BU and yaws. Stigma experiences also differed for adults and children in this setting and these differences should be accounted for in integrated interventions for these skin NTDs. School health programs need to prioritize educating school teachers about skin NTDs and the negative impact of stigma on the wellbeing of children.

### Author summary

Stigma related to Neglected Tropical Diseases (NTDs) constitutes a burden on the social and economic life of affected people, their caregivers and communities. Current evidence on stigma has been limited to a small number of NTDs. The WHO recommends integration approaches to NTDs are therefore understanding stigma and its differential impact is important. We explored stigma experiences, effects and coping strategies simultaneously in BU and yaws in two districts in central Ghana. Findings showed nuances for macro-level stigma associated with BU; individuals were stigmatized by community members; interestingly, this was not noted for yaws. Negative effects of stigma found included psychosocial burden, effects on academic work and straining of social relationships. Some affected individuals dealt with the source of stigma actively avoiding disclosure, selective disclosure, concealment with clothing and confrontation. Emotion-focused strategies such as religious coping and self-isolation were reported; these were noted to aggravate psychosocial problems being experienced by affected people. We propose that, the differences in stigma among people affected by BU and yaws, and the role of age in determining types of stigma identified in this study should be considered in integrated interventions for skin NTDs. School health programs are encouraged to educate teachers about skin NTDs and the negative effects of stigma on students.

## Introduction

Neglected Tropical Diseases (NTDs) are 20 diseases that mainly affect individuals living in poverty in tropical and sub-tropical regions of the world [1–3]. Many NTDs have their major manifestation on the skin (skin NTDs). In Ghana, yaws and Buruli ulcer (BU) are two skin NTDs that constitute important public health challenges; leprosy, scabies and leishmaniasis are also endemic in many communities [4–10]. BU and yaws commonly affect children and are predominant in central and southern Ghana, especially in poor and marginalized communities with limited access to health services, water, and sanitary living conditions [11]. Clinically, BU lesions can present as a nodule, plaque, oedema or ulcer. BU lesions are categorized into 3 groups based on the size as follows: category I- lesions less than 5 cm in diameter, category II- lesions between 5 and 15 cm in diameter and category III- single lesions greater than 15 cm, or multiple or oedematous lesions. Yaws is caused by *Treponema pallidum* subspecies *pertenue* and is characterized by cutaneous lesions (chiefly papilloma or ulcer) during the primary and secondary stages and destructive lesions of the bones in the tertiary stage of the disease [5]. Late detection of BU has been shown to be associated with long term consequences including a mental health burden and reduced quality of life of affected people and their caregivers [12,13].

Several skin NTDs have been associated with stigma within varying social contexts [14,15]. People with NTDs experience stigma through multiple mechanisms. Biologic factors such as physical impairments caused by NTDs mediate stigma. Furthermore, fear of disease transmission and lack of knowledge about a disease are psychological factors that lead to higher levels of stigmatization of affected individuals and their families [16]. Studies of skin NTDs suggest that the diseases may be perceived as symbols of disgrace within some social contexts, inhibiting social participation and leading to socio-economic hardship for affected people [17]. Traditional beliefs in witchcraft, curses, or evil spirits have been linked to the stigma experiences in leprosy, Buruli ulcer [18], yaws [19] and scabies [20]. In podoconiosis, beliefs about the hereditary nature of the disease are associated with stigma reported by affected individuals [21]. There is some data on how stigma may affect health seeking in NTDs [18,22–24]. While stigma was reported to delay care seeking in BU [18], Mulder et al [24] did not find a significant relationship between the stigma and delayed presentation.

In 2020, the World Health Organization (WHO) recommended integrated approaches to the management of NTDs to improve efficiency, cost-effectiveness and sustainability, including activities designed to prevent and manage NTD-related stigma [25,26]. While some research has been conducted to explore the acceptability and patient experiences of integrated care [27], minimal attention has been made to exploring the overlap of stigma in co-endemic NTDs. This evidence may be useful in informing programs about the stigma overlaps and nuances between co-endemic skin NTDs and the general and specialized support that can be provided to individuals affected with different diseases. Additionally, while there is a growing literature on stigma associated with BU in Ghana and other parts of West Africa [28, 29], there has been much less research on experiences of yaws. In two districts of central Ghana, we therefore set out to explore the experiences of stigma, effects and coping strategies among individuals affected by BU and yaws to understand overlaps and disease-specific nuances to help guide integrated interventions.

## Theoretical framework

The sociological construct of macro and micro interaction and Goffman's treatise on stigma served as theoretical frameworks for the study. Macro and micro analysis of social behavior asserts that human behaviour is likely to change in accordance with variations in social context and levels of social interaction [30]. Practically, the study investigated stigma experiences within the individual's private space (micro experiences) and from other community members (macro experiences) as indicated in Fig 1. According to Goffman, stigma discredits and reduces the 'whole' person to a 'tainted' or 'discounted' person, leading to a 'spoil' in identity [31,32]. Consequently, the individual experiences discriminatory treatment. This affects the stigmatized individual either positively or negatively. To manage these experiences, the individual may employ a set of strategies which may positively or adversely impact the situation [32–35].

## Methods

### Ethics statement

Ethical approval was obtained from the institutional review board of the Noguchi Memorial Institute for Medical Research (NMIMR) at the University of Ghana (approval number: FWA00001824) and the London School of Hygiene & Tropical Medicine, UK (approval number: 22604). Full disclosure and explanation of the study procedures was provided to all participants in a language of their choice (English or the local language, Twi). Written informed consent was obtained in English from all participants. For individuals who could not read or

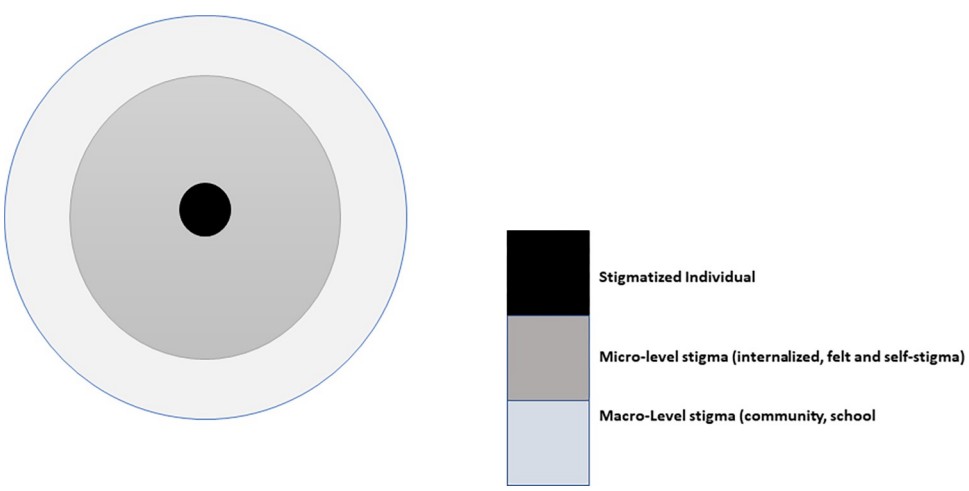

**Fig 1. Conceptualization of levels of stigma along varying levels of social interaction.**

understand English, consenting was done via an interpreter in the presence of a witness of the individual's choice. Following this, persons who could not sign, thumb printed their consent for inclusion in the study. For young children (<18 years), written consent was obtained from parents or legal guardians. Individuals aged 11–17 years provided assent in addition to informed consent obtained from their parents or legal guardians. Study staff received training on how to appropriately conduct interviews. All the study processes were conducted in accordance with the principles guiding research in human subjects as set out in the Declaration of Helsinki [36].

## Study setting

This study was conducted in Atwima Mponua and Wassa Amenfi East districts in central Ghana where skin NTDs including BU and yaws are endemic. Atwima Mponua is located in the south-west of the Ashanti region of Ghana and has a population of 155,254. About 88% of the district's population reside in rural areas. Agriculture (farming, fishing, forestry) constitutes the main economic activity; small-scale mining activities are increasing [37,38]. Twi is the most common spoken language. However, other ethnic groups including Ewe, Bono, Fanti and Krobo, are also present in the district. There has been no formal assessment of the prevalence of yaws and BU in Atwima Mponua. Wassa Amenfi East has a population of 83,478, the majority of which reside in rural communities. Farming is the predominant occupation. The Wassa remain the dominant ethnic group but others such as the Ashanti, Akyem, Sefwi and Nzema reside within the district.

## Study design and participant recruitment

The team at the Kumasi Centre for Collaborative Research in Tropical Medicine (KCCR) operates a clinic for individuals with skin NTDs including Buruli ulcer and yaws in the Wassa Amenfi East district. Health records were reviewed to identify and trace people who were currently receiving treatment or had been previously treated for BU. In 2021, the KCCR collaborated with the district health management team at Atwima Mponua to confirm and manage a yaws outbreak in the district. Individuals who were currently receiving treatment or had been previously treated for BU or yaws from these two settings were contacted for recruitment into the study. Individuals attending the skin NTD clinic at Wassa Amenfi East hospital were contacted in person for recruitment. Participants in Atwima Mponua district were initially

contacted via telephone and the study explained to them. Following the initial telephone discussions, an in-person meeting was held with affected individuals during usual clinic visits in the community or at a nearby health facility. Only individuals who were willing to participate and provide written informed consent were included in the study.

This study utilized face-to-face semi-structured interviews. A convenience sampling technique was employed to identify and select participants for the study. The sample size was intended to generate information on disease specific differences in stigma between BU and yaws. All identified individuals affected by either of the two diseases of interest were included to make the sample representative of the experiences of people with BU and yaws in the study area. Furthermore, data saturation in the open questions was included in the decision making on the sample size. Between June-July 2022, interviews were conducted by female and male study team members (RDT, LO and MNO) in private consulting rooms during usual clinic visits in the community or at a nearby health facility as convenient for participants. Adults and participants 10–17 years were interviewed directly. Interviews of participants 10–17 years old were conducted in the presence of a parent or caregiver. The main caregiver of affected children younger than 10 years was interviewed. The interviews lasted around 30 minutes and were audio recorded. The study's findings have been reported in accordance with the Consolidated criteria for Reporting Qualitative Research (COREQ) checklist (S1 Material).

## Data collection

A semi-structured interview guide covering participant demographics, experiences of macro and micro stigma, effects of stigma and coping strategies was used for data collection. Individuals previously treated for BU or yaws provided additional information on their experiences post-healing. The guide was pretested using mock interviews by the study team on seven affected individuals in Ahafo Ano North, a nearby district also endemic for BU and yaws. No major changes to the study guide were considered necessary. Before data collection, all study team members received training (led by YAA) on administration of the study tool (S2 Material).

## Data analysis

Audio recordings of interviews were transcribed from local languages (Twi and Krobo) to English. Quality control checks were done on sections of audio recordings and transcripts by another team member to ensure accuracy and full transcripts were reviewed by the original interviewer. A coding structure was developed and performed by RDT using QDA Miner Lite version 4 to deductively generate three broad thematic areas based on the concept of macro-micro analysis and Goffman's stigma treatise, exploring participant stigma experiences, the effects and coping strategies in stigma management. Macro-level stigma included stigma perpetrated by other community members towards affected individuals. Micro experiences included accounts of internalized, self, felt and anticipated stigma. Effects of stigma were refined to identify subthemes which were also further analyzed and separated into problem-focused coping strategies (use of religion, concealment with clothing, information management and selective disclosure) and emotion-focused coping strategies (self-isolation and use of religion).

## Results

### Disease characteristics of study participants

The predominant lesion type among BU participants was an ulcer and most had category III disease. Among participants with yaws, the most common form was an ulcer and most had healed lesions at the time of assessment (Table 1).

**Table 1. Lesion characteristics of Buruli ulcer and yaws participants.**

| Disease | | | Frequency |
|---|---|---|---|
| Buruli ulcer (BU), n = 15 | Lesion type | Nodule | 3 |
| | | Oedema | 2 |
| | | Ulcer | 10 |
| | Lesion category* | Category I | 2 |
| | | Category II | 3 |
| | | Category III | 10 |
| | Healing status | Healed | 8 |
| | | Not healed | 7 |
| Yaws, n = 16 | Lesion type | Papilloma | 5 |
| | | Ulcer | 11 |
| | Healing status | Healed | 11 |
| | | Not healed | 5 |

*Category I: small (<5 cm diameter) lesions; Category II: medium size lesions (5–15 cm); Category III: large (>15 cm) lesions, multiple lesions or lesions at critical sites (e.g genitals, perineum, eyes)

## Participant characteristics

Thirty-one current and previously affected individuals participated in the study, 15 with BU and 16 with yaws. The majority of study participants were below 15 years (n = 25, 81%). A slightly higher number of participants were males (n = 18, 58%). Twelve (38%) of study participants were currently on treatment while nineteen (61%) were former patients at the time of data collection. The socio-demographic characteristics of participants are presented in Table 2.

## Experiences of Stigma

**Macro-level stigma.** Community members, school mates and school staff were reported to stigmatize individuals currently or previously affected by BU (n = 14, 93.3%). Individuals who had non-ulcerated forms of BU reported less stigma compared to those who had ulcers. The macro-level stigma associated with yaws was reported by 5/16 (31.3%) of affected individuals and their caregivers; this was out of concern of contagion by schoolteachers who considered sending affected children home as a means of protecting other school children. Common forms of stigma experienced by individuals affected by BU included name-calling and staring. Younger participants (15 years and below) reported teasing and name-calling by peers, playmates and occasionally siblings using derogatory words such as *apakye* (bad limb), 'torchlight' (referring to BU ulcer) or referencing 'smelly wounds' for large and undressed wounds in particular:

"...*when I leave it [ulcer] open without dressing it, some people will see me and say, look at the torch [ulcer]... it made me feel very sad, there were even times that I cry when they refer to my wound as torch."* [11-year-old female, formerly affected with BU, 001].

Older participants and caregivers mainly discussed the pain of dealing with public staring:

"*They [community members] can stare at me because I think they wonder why such an old person like me has a bandage on her leg. Especially, when I walk around with my stick, I bow down my head, they can stare at you for such a long time that I don't even know how to walk again [confused walk around in public]"* [60-year-old female, formerly affected with BU, 004].

**Table 2. Socio-demographic characteristics of participants.**

| Variable | Buruli ulcer (n = 15) | Yaws (n = 16) | Total (n = 31) |
|---|---|---|---|
| **Age (years)** | | | |
| 1–15 | 10* | 15# | 25 |
| 16–30 | 3 | 1 | 4 |
| 46–60 | 1 | 0 | 1 |
| ≥ 61 | 1 | 0 | 1 |
| **Sex** | | | |
| Male | 6 | 12 | 18 |
| Female | 9 | 4 | 13 |
| **Educational status** | | | |
| None | 3 | 2 | 5 |
| Primary | 8 | 14 | 22 |
| Secondary | 3 | 0 | 3 |
| Tertiary | 1 | 0 | 1 |
| **Religion** | | | |
| Christianity | 10 | 9 | 19 |
| Islam | 3 | 5 | 8 |
| Traditional | 2 | 2 | 4 |
| **Occupation** | | | |
| Self-employed | 4 | 1 | 5 |
| Unemployed | 3 | 0 | 3 |
| Students | 8 | 15 | 23 |
| **Ethnicity** | | | |
| Akan | 9 | 10 | 19 |
| Mole-Dagbani | 4 | 5 | 9 |
| Others | 2 | 1 | 3 |
| **District** | | | |
| Atwima Mponua | 0 | 14 | 14 |
| Wassa Amenfi East | 15 | 2 | 17 |

6 participants with BU were ≤ 9 years*

7 participants with yaws were ≤ 9 years#

Avoidance or exclusion from group activities was noted in both BU and yaws, especially among children, although different motives were described. In yaws, exclusion was interpreted by affected children and caregivers as motivated by a concern for the wellbeing of participants:

"*When I get up to go and play, they [other children] mostly fear the sore will be bruised so they don't allow me to play*" [7-year-old male, formerly affected with yaws, 001].

Meanwhile, motives for excluding BU patients were interpreted by affected people to be unfriendly and fed by a fear of contagion:

"*Before I had BU, they [classmates] were close to me, but after I had the condition, they were afraid that they will get infected if they come closer to me. When I go to school, all the pupils do not want to get closer to me*". [11-year-old female formerly affected with BU, 003].

"*After I had BU, they [friends] do not seem friendly anymore. They behave as if my wound is smelling and avoid me when I go near them. I sit at my desk in the class alone. When it is closing time, I come to my house alone*" [14-year-old female, currently affected with category III BU ulcer, 001].

Moreover, exclusionary behaviour was enacted not only by other students, but by some teachers and school food vendors. Participants explained that these actions were justified to them by school staff in terms of infection control to protect other students and staff. For yaws, this tended to involve asking parents to keep affected students at home while symptomatic, while for BU this involved enacting sometimes elaborate procedures such as asking students to bring their own chairs and bowls from home or insisting students participate in class discussions only through written messages, which also had the effect of drawing attention to children's diseases:

"...*before the community volunteer came to take me to the hospital, it was very bad. [...] They [the teachers] wanted me to start bringing my own chair to school [... then] the headmaster asked me later to take my chair home and stay in the house till I am healed. [...] one of my teachers too, [...] when I know the answer to solve the question on the black board, she makes me write the answer on a paper and askes someone closer to me, my relative, to bring her the paper. They do not allow me to get close to the black board.*" [caregiver of 9-year-old female, currently affected with category III BU ulcer, 014].

"...*When I go to the canteen to buy food, I have to send my own bowl [through a friend]. Unless I do that, they* [food vendors] *will never sell the food for me with their own bowl.*" [11-year-old female formerly affected with BU, 003].

**Micro-level stigma experiences.** Micro-level stigma was observed for all individuals with BU (n = 15, 100%). Most people affected by BU internalized being poorly treated and withdrew from social interaction in anticipation of similar treatment from other people. Many participants who expressed this form of self-stigma had more advanced disease such as large ulcers, though anticipatory stigma was also present in people in the early stages. A participant, who had a category II BU ulcer (size between 5 cm to 15 cm) on her thigh expressed how she became self-conscious and reluctant to reveal her affected body part to other people;

"*I was shy that I had such a wound on my leg*" [18-year-old female, formerly affected with BU, 002]

Another person previously affected by a category III ulcer BU (size >15 cm) reported she had stopped sharing cooked food with neighbors and, even though she had finished hospital treatment, she still prefers to only share uncooked food items as she feared cooked food may be rejected due to her contact with the food items;

"*Well, because I knew my own problem* [BU], *I didn't even try that [giving out cooked food to neighbors and friends]. Some people might take it* [the cooked food] *but they won't eat it and pour it away*" [60-year-old female, formerly affected with BU 004].

For yaws, parents perceived that affected children did not internalize stigma, but caregivers themselves exhibited high levels of courtesy stigma (n = 9, 56.3%) because of popular associations between yaws and poor personal hygiene:

"*As for the children, they do not care. They are able to play with their friends. They go to school every day. They are not shy. It is me who is worried, some people may say that I am not taking good care of the child and not being attentive to them when they are sick*" [mother of 6-year-old male, currently affected with yaws, 008].

"*As an adult, it would have been difficult to go to the mosque or the market if the sores are on visible parts of the skin. It is embarrassing. [. . .] You will know that they* (other community members) *are all gossiping about you but you wouldn't be able to do anything about it*" [mother of 9-year-old male, currently affected with yaws, 007].

Many mothers declared that they would be very unlikely to marry a man who had yaws as the ulcers or scars may compromise physical attraction. Some reservations were also because of being uncomfortable with community perceptions of poor personal hygiene as a causative factor.

## Effects of stigmatizing experiences

Stigma associated with BU appeared to have several negative effects including imparting a psychosocial burden, interfering with young people's academic work and straining relationships. In both BU and yaws, stigma appeared to motivate healthcare-seeking and use.

**Psychosocial burden of BU.** Significant feelings of anxiety and sadness were reported. Younger BU participants particularly reported feeling sad due to negative interactions with friends and playmates. Older adolescents also described feeling hurt, crying and withdrawing from group activities after stigmatizing experiences:

"*One of my friends, while I was arguing with her, she said, 'that is why you have [. . .] your big sore on your leg' [. . .] she did that when we were in class, our classmates were there [. . .] I cried. I was really hurt. Because of what happened, I did not feel comfortable to get close to my classmates anymore*" [18-year-old female, formerly affected with BU, 002].

**Academic effects.** Some participants were compelled to stay away from school, especially when the disease was at its worst. This was largely due to pain associated with the disease but also the experience or fear of stigma from others. Students with BU reported that worrying about how others treated them was a distraction during classes and affected their work, though this improved as wounds began to heal:

"*Well, my learning was affected and I also did not want to go to the school anymore so that they will keep telling me to go home, go home. And I kept asking myself, when at all will my wound heal? There are times I think about this so much that even when someone calls me, I cannot hear the person unless the person touches me*" [11-year-old female, formerly affected with BU, 003].

"*I used to think a lot that how could my own best friend, whom I shared everything with, and go everywhere with, do that to me? So, I can be thinking about this during classes [. . .] I was second in class before I got the BU that was in form one. But after, I got the 8th position in form 2, now in form 3, I was 5th [. . .] The most recent one we had after my leg got healed, I was 3rd in the class*" [18-year-old female, formerly affected with BU, 002].

The desire to conceal disease from peers in anticipation of stigma led an adolescent to change her school arrangements from boarding to become a day-student.

**Effect on social relationships.**    The quality of social relationships was affected after participants affected by BU experienced or anticipated stigma from people that they related with. Many participants related with other people differently:

"*We talk, but our relationship is not like it was at first. . .I still talk with some people, but it is not like previously that I will gather with my friends and we be chatting with each other. It is not like that anymore*" [17-year-old female, formerly affected with BU, 012].

"*Hmmm, because of what happened, I did not feel comfortable to get close to my classmates anymore*" [18-year-old female, formerly affected with BU, 002].

Even though many friendships were affected after experiences of stigma, family and intimate partners, both among the married and unmarried remained supportive of affected partners.

"*He [participant's husband] was very kind to me. He stopped me from performing some of the house chores, so he and the children were doing most of the things in the house. He was so good to me. . .some of my family members also travelled all the way to come and stay with us for a while when they heard that the disease which has affected my children and even brother in the past has also affected me.* [61-year-old female, formerly affected with BU, 006]**.**

"*He* (participant's intimate partner) *asked me to bring the condition to the hospital, but he still gets close to me, nothing has changed*". [24-year-old female, currently affected with BU, 009].

For participants with yaws, there was no observed effect of stigma on social relationships.
**Effect on health-seeking behavior.**    Health-seeking decisions for the diseases appeared to be varied, and reflected differences in individual meanings and interpretation associated with the diseases:

". . .*everybody [community members] says their own mind about it [BU]. They keep advising that you send the disease to so many places [varying care alternatives]. Some will even tell you that the care I am getting in the hospital is not good and show us some different places to go to*" [11-year-old female, formerly affected with BU, 001].

Treatment pathways sought were likely to change based on affected people and their caregivers' experiences and perceptions of treatment effectiveness, which were seen to vary even within families:

"*There was a man (traditional healer) who was staying around my house, he was the one who was taking care of the wound. He told me that he will cut out the slough on the wound after some time. That was why we stopped going there. [. . .] my sister said she will not sit idle and allow the man to cut into my wound so she carried me on her back one day when no one was in the house* [patient's mother, did not agree with the decision to change treatment modality at the time], *because I could not walk at the time and sent me to the hospital*". [11-year-old female, formerly affected with BU, 003].

It was found that internalized and anticipated stigma may motivate health-seeking by both BU and yaws participants and adherence to medication and home-based dressing protocols for BU:

"It [yaws] *is embarrassing, so as an adult, you will do all you can to get a cure so you will be free and be able to go out.* [mother of 9-year-old male, currently affected with yaws, 007].

"*Some* [neighbors] *didn't even want my son to come play around them, but since he has completed with the medicine, I dress the wound every three days, I know that it will heal soon even though it is slow right now.*" [father of 8-year-old male, currently affected with category III BU, 008].

## Coping strategies

Coping strategies to manage stigma included problem and emotion-focused strategies. Problem-focused strategies included concealment with clothing, selective disease disclosure, confrontation and education. Emotion-focused strategies included finding comfort in religion and self-isolation.

**Concealment with clothing.** Clothing was widely used to conceal lesions and scars to limit or prevent other people's reactions. Many BU participants, both young and old, used clothing not for adaptive purposes (managing physical disabilities or aiding movement), but to limit attention being drawn to affected body parts and in some cases to protect the surface of lesions from contamination with dirt:

". . . *now, I wear longer clothes to cover the wound especially when I am sent to the market or I am going out of the house. Sometimes in the house, I try to wear dresses that cover the wound so that people who come to the house will stop asking me what is wrong with my leg, yes so there is a little change now*" [14-year-old female, currently affected with BU, category 2 ulcer, 001].

Clothing remained useful to some in concealing scars even after BU patients healed:

"*When people see it* [BU scar], *they will be asking, so when will this scar heal completely and disappear from your skin*? *Even some old people keep asking me the same thing*" [11-year-old female, formerly affected with BU, 003].

Participants with yaws did not feel they needed to use clothing to conceal lesions, as they were less concerned about others seeing them. Participants felt comfortable having their lesions or scars exposed regardless of their social settings; this may reflect the differences in nature of early yaws and BU scars.

"*Other people may have children who also have sores on their legs so they cannot talk about my children in a bad way. Also, for children it is normal for them to get hurt as they play*" [mother of 6-year-old male, currently affected with yaws, 008].

**Selective disease disclosure.** Individuals with yaws and their caregivers openly shared information about the condition with friends, wider community and school mates. However, BU participants reported that they carefully selected who they disclosed information to about their condition. Even in ulcerated cases where wound dressing materials were visible, many participants were unwilling to disclose the cause of the ulcers to be BU:

"*For my school mates, when they ask, I tell them my arm is paining me. I never mentioned the type of disease to them [. . .] I did not want them to behave like how my friends from my neighborhood were behaving*" [15-year-old male, formerly affected with BU, 005].

**Confrontation and education coping strategy.** Some younger participants affected by BU directly challenged people who perpetuated blatant or subtle stigmatizing behaviors such as discrimination and avoidance towards them, though these attempts were not always successful at changing opinions. A participant reported how she confronted and tried educating her playmates, although inaccurately, about the condition to dispel their fears of contagion, which was manifested in excluding her from group activities:

"*I ask them, why are you telling me to leave? You will only get BU when it is in your blood. The Community Volunteer told me that, it is not everybody who is susceptible to get BU. He told me that the organisms that cause the disease can sometimes be found in dirty bath water and near water bodies, for instance associating myself with rivers or streams. But even after I explained to them, they will still ask me to leave them alone.*" [11-year-old female, formerly affected with BU, 003].

**Comfort in religion.** An overwhelming majority of BU participants found comfort in religion to manage the distress associated with stigma. When asked what they did after experiencing stigma, most mentioned praying, attending church, or performing other religious activities. Many stated their only hope of healing was God:

"*Oh, I just rely on God. I have nothing except for God . . .Oh, I sit, I pray in my head, God heal me, heal me. God wants work to do so if you don't give him work, he won't save you. So, I say, God heal me, so I give God work and he is working* [60-year-old, female, formerly affected with BU, 004].

**Isolation and behavioral change.** Both BU and yaws participants reported self-isolation and behavioral change after onset of disease. Individuals with yaws typically did this due to associated pain but those with BU withdrew in response to stigma. BU affected individuals reported that they tended to avoid other people and preferred to be by themselves. Some participants who were extroverts previously, mentioned that they had become reserved because of how they were treated when they had the disease. Even after healing, one participant mentioned that she is more withdrawn:

"*I am quiet now and will continue to be*" [18-year-old female, formerly affected with BU, 002].

## Discussion

In this study, we show that the types and levels of stigma vary for BU and yaws. Further, stigma experiences differ among children and adults in both BU and yaws. Stigma experiences were perceived in this study to contribute to negative outcomes. Participants who used religious coping tended to report higher psychosocial suffering as compared to those who coped using active mechanisms such as confrontation.

Whereas there is growing evidence of stigma related to BU [28,29], limited attention has been given to potential stigma associated with yaws. A recent study in the Philippines [39], showed that adults with yaws in Mindanao but not Luzon experienced stigma because of positive serological tests which are also positive in individuals with syphilis. The authors concluded "a positive test is highly stigmatizing because of the association with promiscuity and fear of contagion. Affected individuals were excluded from employment and income opportunities. In the same study, affected children reported being bullied in school because of the condition.

We found that children affected by yaws were excluded from play activities by peers due to concern of causing further harm or trauma through rigorous play. The varying social contexts and differences in knowledge about the disease may account for the observed experiences in Ghana and the Philippines. In BU, participants in this study faced considerable restrictions in societal participation; this is similar to previous reports from other parts of Ghana [40]. Participants who had non-ulcerative forms of BU reported less exclusion compared to participants who had large ulcers which took longer time to heal. This is similar to a previous scoping review on cutaneous leishmaniasis where lesion type, severity and scarring were considered important in both social and self-stigma [41].

Micro-level stigma was common in both BU and yaws in this study, contributing to self-isolation and psychosocial burden. This is consistent with findings from Indonesia and Ethiopia where individuals affected by leprosy and podoconiosis (a non-infectious lymphoedema) reported internal self-stigma [42,43].

In another study in Ghana, people affected with BU (with mean age 20 years) were found to be hindered from functioning as leaders, felt ashamed or embarrassed and thought less of themselves [40]. These individuals felt avoided by community members and had difficulty in finding marriage partners. In our study, individuals affected by BU reported psychosocial burden, including anxiety, sadness and feelings of embarrassment which were more prominent among persons with more severe clinical disease. Individuals with yaws were sent home by school authorities for fear of transferring the disease to other school children; this finding differs from the Philippines where affected children suffered bullying which contributed to school absenteeism [39]. The difference may be related to varying socio-cultural interpretation of disease in the two countries. We propose that as part of school health education programmes, teachers should be educated and trained to identify, refer and offer psychosocial support to students affected with skin NTDs. Skin NTDs have previously been noted to promote a decline in religious and neighborhood relationships [44–46] and similar impacts were seen in the current study.

We found that for participants with yaws, there was no effect of stigma on social relationships. A possible explanation for this observation might be the fact that yaws lesions tend to be smaller and less disfiguring than BU ulcers and so induce less stigma and be socially acceptable. This is supported by the finding that non-ulcerated forms of BU were associated with less stigma than large ulcers. Additionally, yaws lesions may not have the odorous smell that is sometimes present in large BU lesions especially when they are secondarily infected [47].

Different coping strategies for managing stigma have been recorded in the literature. In this study, concealment of wounds and scars with clothing was extensively used in managing stigma associated with BU, as recorded in other chronic conditions such as podoconiosis and psoriasis [48,49]. This is consistent with Goffman's theory, which proposes that stigmatized individuals often attempt to appear 'normal' by concealing the source of the stigma [31]. This has also been seen in Nigeria where individuals suffering from lymphatic filariasis reported covering the affected legs with clothing to gain community acceptance and engage in social activities [50]. Another way of coping for some affected individuals was keeping the disease secret, only disclosing to selected people such as family members and close friends. Individuals who were unable to conceal lesions from the public, withheld or gave vague or incorrect information about the lesion when others enquired.

Consistent with findings on stigma associated with HIV and leprosy, religion was used by BU affected individuals to manage stigma [51–53]. In Indonesia, religion positively helped individuals affected by leprosy cope with the disease as it gave meaning (including punishment for past sins, challenge or test of faith by an object of worship) to the disease [52]. Unlike Indonesia however, affected individuals in the present study did not regard their symptoms as a

punishment or test of faith; the observed difference may reflect disease-specific characteristics or the disparities in socio-cultural contexts between Ghana and Indonesia.

Self-isolation was a coping strategy by BU participants in this study and this is consistent with studies of HIV, leprosy and cutaneous leishmaniasis [54–56]. However, avoidance strategies are commonly associated with low self-esteem, increased levels of anxiety, depression, reduced quality of life, impaired psychological and mental well-being and suicidal ideation [41,57]. Consistent with this, individuals with severe disease in the current study who used self-isolation for coping, reported an increased psychosocial burden, including worry, sadness and distraction in school.

Stigma and supernatural explanations of disease causation (such as witchcraft and curses) are usually linked to diseases which are difficult to cure or chronic and in which causation is difficult to explain [20]. However, in a previous study in Ghana, some 13% of participants believed that yaws was caused by supernatural forces such as witchcraft, curses or a punishment from god and this belief was not associated with gender, religion or ethnicity [19]. In another study, Buruli ulcer was associated with supernatural causes by persons in endemic communities [18,58]. The similarity of cultural, religious, ethnic backgrounds and their impact on belief systems may account for the observed findings in the current study but we are, however, uncertain as we did not explore the impact of local beliefs on the causation and aetiology of BU or yaws in this study.

Some evidence indicate a link between stigma and reduced health seeking [59, 60]. We found that the fear of stigma and experiencing stigma motivated parents and guardians to seek and adhere to treatment, whether biomedical, traditional or religious interventions for affected individuals. Other evidence supports this effect of stigma which motivates health seeking and treatment adherence, mediated by the individual's confidence in the treatment sought to cure the disease, and hence, diminish the associated stigma [61]. However, further research is needed to examine how stigma associated with BU or yaws and other disorders may interact in programmes where services to diagnose and manage these conditions are integrated within healthcare facilities.

## Study strengths and limitations

Our study adds new insight to the skin NTD stigma literature in terms of its large focus on children and their caregivers and a comparative focus on BU and yaws, bringing to light the disease-specific similarities and differences which are important to inform integrated interventions within the context. The study included all the identified individuals with the disease and so the sample is representative of the experiences of people with BU and yaws in the study area. However, the study was limited in not exploring gender or cultural differences and intersectionality including local beliefs on disease causation and aetiology, which may give further insight to understanding stigma perpetrated against affected individuals. Differences in disease distribution and reporting practices by health facilities also influenced our recruitment. Most people with yaws came from Atwima Mponua while everyone with BU came from Wassa Amenfi East, but we believe there are enough shared sociocultural characteristics that our analysis is reflective of a broader central Ghanaian social context.

## Conclusion

Stigma is experienced by individuals with BU and yaws. However, we noted variation in the types and levels of stigma. Comparatively, yaws was considered more socially acceptable, and peers were willing to associate with affected individuals, unlike in BU where enacted stigma such as discrimination, teasing and name-calling perpetuated by peers and adults within the

school setting were common. These stigma experiences resulted in psychosocial burden, underachievement in school and strained social relationships for affected individuals. Some coping strategies used in the management of stigma contributed further to worsening of experiences.

We propose that the differences in stigma among people affected by BU and yaws, and the role of age in determining types of stigma, should be considered in the design and implementation of integrated interventions for skin NTDs. Interventions for school health programmes should incorporate teacher education about skin NTDs affecting their communities to avoid misinformation and unnecessary, stigmatizing and deleterious practices such as student exclusion from schools. Teachers are a potentially important human resource in addressing negative impact of stigma on the wellbeing of affected students and promoting the importance of treating affected people with respect.

## Supporting information

**S1 Material. Consolidated criteria for Reporting Qualitative research (COREQ) Checklist.** (PDF)

**S2 Material. Interview guide for stigma study.** (PDF)

## Acknowledgments

We wish to thank the individuals and communities for their participation in the work of the Skin Health Africa Research Programme.

## Additional members of the SHARP Working Group

Olivia Dornu, Philemon Boasiako Antwi, Eric Koka, Collins Ahorlu, Daniel Okyere, Edmund Kwaku Ocloo, Adwoa Asante-Poku, Endalamaw Gadisa, Mirgissa Kaba, Saba Lambert, Esther Amon

## Author Contributions

**Conceptualization:** Ruth Dede Tuwor, Tara B. Mtuy, Yaw Ampem Amoako, Jonathan Mensah Dapaah, Richard Odame Phillips, Jennifer Palmer.

**Data curation:** Ruth Dede Tuwor, Abigail Agbanyo.

**Formal analysis:** Ruth Dede Tuwor, Yaw Ampem Amoako, Lucy Owusu.

**Investigation:** Ruth Dede Tuwor, Lucy Owusu, Michael Ntiamoah Oppong, Abigail Agbanyo, Bernadette Agbavor.

**Methodology:** Ruth Dede Tuwor, Tara B. Mtuy, Yaw Ampem Amoako, Jonathan Mensah Dapaah, Richard Odame Phillips, Jennifer Palmer.

**Project administration:** Yaw Ampem Amoako.

**Resources:** Michael Marks, Stephen L. Walker, Dorothy Yeboah-Manu, Rachel L. Pullan, Richard Odame Phillips.

**Supervision:** Tara B. Mtuy, Yaw Ampem Amoako, Jonathan Mensah Dapaah, Richard Odame Phillips, Jennifer Palmer.

**Visualization:** Michael Ntiamoah Oppong, Bernadette Agbavor, Michael Marks, Stephen L. Walker, Dorothy Yeboah-Manu, Rachel L. Pullan.

**Writing – original draft:** Ruth Dede Tuwor, Tara B. Mtuy, Yaw Ampem Amoako.

**Writing – review & editing:** Ruth Dede Tuwor, Tara B. Mtuy, Yaw Ampem Amoako, Lucy Owusu, Michael Ntiamoah Oppong, Abigail Agbanyo, Bernadette Agbavor, Michael Marks, Stephen L. Walker, Dorothy Yeboah-Manu, Rachel L. Pullan, Jonathan Mensah Dapaah, Richard Odame Phillips, Jennifer Palmer.

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
