## [Decision Letter · Decision Letter 0]

18 Dec 2023

Dear Amoako,

Thank you very much for submitting your manuscript "Stigma experiences, effects and coping among individuals affected by Buruli ulcer and yaws in Ghana" for consideration at PLOS Neglected Tropical Diseases. As with all papers reviewed by the journal, your manuscript was reviewed by members of the editorial board and by several independent reviewers. In light of the reviews (below this email), we would like to invite the resubmission of a significantly-revised version that takes into account the reviewers' comments. 

We cannot make any decision about publication until we have seen the revised manuscript and your response to the reviewers' comments. Your revised manuscript is also likely to be sent to reviewers for further evaluation.

Sincerely,

Mathieu Picardeau

Section Editor

Reviewer's Responses to Questions

**Key Review Criteria Required for Acceptance?**

**Methods**

-Are the objectives of the study clearly articulated with a clear testable hypothesis stated?

-Is the study design appropriate to address the stated objectives?

-Is the population clearly described and appropriate for the hypothesis being tested?

-Is the sample size sufficient to ensure adequate power to address the hypothesis being tested?

-Were correct statistical analysis used to support conclusions?

-Are there concerns about ethical or regulatory requirements being met?

Reviewer #1: (No Response)

Reviewer #2: Are the objectives of the study clearly articulated with a clear testable hypothesis stated? YES

-Is the study design appropriate to address the stated objectives? NO

-Is the population clearly described and appropriate for the hypothesis being tested? NO

-Is the sample size sufficient to ensure adequate power to address the hypothesis being tested? NO

-Were correct statistical analysis used to support conclusions? YES

-Are there concerns about ethical or regulatory requirements being met?YES

Reviewer #3: The authors have not clearly stated the objectives of the study or clearly articulated research questions derived from their understanding of the problem (either from literature or their observation), which they seek to explore. While stating that this paper will attempts to address gaps, they have identified by exploring stigma experiences of people currently or previously affected by BU and yaws, the authors are requested to be clearer on what they sought to explore in the introduction. Suggestion to organize these objectives/research questions to align with how the results have been presented: experiences of stigma, effects, and coping strategies.

Study design: The qualitative study design using face-to-face interviews employed is appropriate. Although the authors state that they used a purposive sampling technique, they are not clear on the specific characteristics/attributes they based their sampling on. Why then is this not convenience sampling?

Sample size: More detail is required here. How many cases did they anticipate identifying? How many did they expect to interview? What was the minimum sample size required for meaningful analysis? As this is not a key-informant interview and the authors will be generalizing their findings on patient experiences, this is a significant omission.

Methodology: The authors stress the interviews were carried out by female study team members. Is there a special reason why? Is there a possibility of bias limiting the sex of interviewers to one sex?

Ethical/Regulatory concerns: The authors state that health facility records were reviewed to identify and trace people who were currently receiving treatment or had been previously treated for BU or yaws. Who carried out this review? The health facilities have no right to divulge someone’s disease status to a researcher; this is unethical. One suggested approach was for health workers to have obtained some form of consent from identified patients before could be contacted by researchers. If the facilities concerned are Ghana Health Service (GHS) facilities, it would constitute a significant omission on the part of authors not to have sought clearance with the GHS Ethics Research Committee. Was there also regulatory approval by the relevant agency concerned for the health facilities? This is not stated.

**Results**

-Does the analysis presented match the analysis plan?

-Are the results clearly and completely presented?

-Are the figures (Tables, Images) of sufficient quality for clarity?

Reviewer #1: (No Response)

Reviewer #2: Does the analysis presented match the analysis plan? YES

-Are the results clearly and completely presented? NO

-Are the figures (Tables, Images) of sufficient quality for clarity? NO

Reviewer #3: Results are clearly presented. Information in tables has been clearly presented.

A minor point is to ensure tables (e.g. Table 2) do not go across pages in the final print.

**Conclusions**

-Are the conclusions supported by the data presented?

-Are the limitations of analysis clearly described?

-Do the authors discuss how these data can be helpful to advance our understanding of the topic under study?

-Is public health relevance addressed?

Reviewer #1: (No Response)

Reviewer #2: Are the conclusions supported by the data presented? No

-Are the limitations of analysis clearly described?no

-Do the authors discuss how these data can be helpful to advance our understanding of the topic under study? yes

-Is public health relevance addressed? not cleary

Reviewer #3: The conclusions generally support the data presented (while noting earlier concerns about generalizing without adequate sample size and the possibility of bias introduced by limiting sex of interviewers).

The limitations of analysis have also been clearly described.

The authors have also discussed how their findings can advance the understanding of this topic and what public health actions may be required-

**Editorial and Data Presentation Modifications?**

Reviewer #1: (No Response)

Reviewer #2: Ruth Dede Tuwor and co-authors reported stigma experiences, effects and coping among individuals affected by Buruli ulcer and yaws in Ghana.

The study addresses an important psychological aspect in the management of skin NTDs, which is vital to shaping national policies, and thus contributing to skin Health for all.

That notwithstanding, the study has some issues that limit the interest of this paper.

1-In the introduction, there are some missing data to support the arguments and also the use of repeated statements. In particular the different stigma characteristics and the status reports on the two skin NTDs which are the subject of this work. Speaking of yaws and BU specifically, the studies of Doftas et al (2022, https://doi.org/10.1186/s41182-022-00433-4), Azubuike et al (2023, https://doi.org/10.1093 /inthealth/ihad090) and Nwafor et al even though cited (doi: 10.4314/ahs.v19i2.34) are guidelines. More broadly, Koschorke et al. (2022, https://doi.org/10.3389/fitd.2022.808955) provide a significant overview.

2-in the study setting , authors described population, the most common language and ethnics groups in the Atwima Mpouna and Wasa Amenfi East district. But there is no mention of the prevalence of BU and yaws; Authors should add this information

3-the authors chose a female team for the interviews. Perceptions/attitudes are sometimes gender dependent depending on cultures. As a result, the creation of an exclusively female team can cause bias in the responses provided . Authors should explain this choice of the investigation team.

3-There is missing information on the sample size calculation. Authors should give more explanation on the study sample size considering the disease prevalence, type of lesion age and sexe. The sample size in the study seems to biais the results.

Reviewer #3: As noted earlier, tables should not go across pages in the final print.

**Summary and General Comments**

Reviewer #1: This is a very interesting study presenting stigma of yaws and BU in rural areas of Ghana and brings to light new and interesting spects of stigma suffered by those infected with BU and yaws.

I would address the fact that the number or percentage of participants reporting the different levels of stigma is not included in the results, therefore we dont know if the stigma reported is something only one or most participants reported.

Its interesting that almost no stigma in yaws has been reported, when most studies in yaws report stigma. ( see the study in yaws stigma in Ghana by Marks et al 2017). This should be addressed in the discussion as it is an unexpected outcome. 

It would also be beneficial to include a section with the limitations of the study addressing issues such as the small sample size or the sampling methodology.

Reviewer #2: Given the endemicity of the both skin NTDs in the study sites, sample size constitutes a bias in drawing conclusions. The type of skin NTD and the phase of the disease (active infection, just healed, & former patient) also constitute a another bias in terms of provided responses. Participants' responses may vary depending on whether the interviewee has the illness, under treatment or is cured. This work requires restructuring to address study design shortcomings. The conclusion of this work should be carried out in subgroup analysis (i.e. sick vs. just recovered, former patient vs. current patient, under treatment, etc.).

Based on available data, Ghana bears much of the global yaws burden for example. How can the authors explain so much stigma in a population where a treatable disease is prevalent?

Reviewer #3: A very relevant topic and well written paper. Serious concerns, however, remain with the methodology and ethical/regulatory issues.

PLOS authors have the option to publish the peer review history of their article (what does this mean?). If published, this will include your full peer review and any attached files.

Reviewer #1: Yes: Camila González-Beiras

Reviewer #2: No

Reviewer #3: No
---

## [Decision Letter · Decision Letter 1]

20 Feb 2024

Dear Amoako,

Thank you very much for submitting your manuscript "Stigma experiences, effects and coping among individuals affected by Buruli ulcer and yaws in Ghana" for consideration at PLOS Neglected Tropical Diseases. As with all papers reviewed by the journal, your manuscript was reviewed by members of the editorial board and by several independent reviewers. In light of the reviews (below this email), we would like to invite the resubmission of a significantly-revised version that takes into account the reviewers' comments. 

We cannot make any decision about publication until we have seen the revised manuscript and your response to the reviewers' comments. Your revised manuscript is also likely to be sent to reviewers for further evaluation.

Sincerely,

Elsio A Wunder Jr, DVM, Ph.D.

Section Editor

Reviewer's Responses to Questions

**Key Review Criteria Required for Acceptance?**

**Methods**

-Are the objectives of the study clearly articulated with a clear testable hypothesis stated?

-Is the study design appropriate to address the stated objectives?

-Is the population clearly described and appropriate for the hypothesis being tested?

-Is the sample size sufficient to ensure adequate power to address the hypothesis being tested?

-Were correct statistical analysis used to support conclusions?

-Are there concerns about ethical or regulatory requirements being met?

Reviewer #4: - The objectives of the study are clearly articulated, although there are gaps in the introduction.

- No major remarks to add regarding the study design, population description, sample size or statistical analysis.

- Ethical concerns are present. Please refer to the point-by-point discussion.

Point-by-point:

- Lines 124-125: “Importantly, there are very few data on how skin NTD stigma affects healthcare- seeking.” Authors are ignoring several pieces of literature, for instance, https://doi.org/10.4269/ajtmh.2002.67.207, https://doi.org/10.1016/j.trstmh.2008.05.026, https://doi.org/10.1093/inthealth/ihv071, doi: 10.1371/journal.pntd.0000237, https://doi.org/10.1371/journal.pntd.0008030, etc.

- Lines 129-132: authors state that "there has been little research on how programs should address the variation or overlap in stigma experiences for different co-endemic skin NTDs", but this formulation raises several questions:

 o It does not necessarily match their objective or even the discussion of their results, as they only briefly mention the practical implications of their results.

 o "varies between conditions and cultures." Was this at any point addressed? If not, why do the authors want to raise this issue?

 o Also, taking as an example, https://doi.org/10.1371/journal.pntd.0008030. Maybe it would be worth revisiting the literature?

- Lines 132-136: if integration of stigma toward NTD’s was the goal, wouldn't a systematic review of the literature serve that purpose better? If note, why not select also/instead patients with Leprosy or Leishmaniosis? What would you expect to change in your results?

- Lines 183-192: the team at KCCR was said collaborate with the district health management team at Atwima Mponua “to confirm and manage a yaws outbreak in the district”. However, nowhere in the text is revealed its link to BU cases. How did they get access to the list of patients with BU?

- Lines 196-198: “The sample size was intended to both generate information on geographical variation and disease specific differences between BU and yaws.” This is not accurate, since at any time in the manuscript authors address geographical variations and by “disease specific differences” are the authors talking about differences in stigma? This should be clearer. Additionally, this is an exploratory qualitative study. What is really necessary regarding to population sampling is to include information on how many patients were asked to participate in the study, how many refused, and if geographic variations are relevant, information on the provenience of subjects.

- Lines 217-218: “The guide was pretested in Ahafo Ano North, a nearby district also endemic for BU and yaws.” Is this published? If not, how has it been pretested? Who pretested? Using what population sample?

- Line 228: “Other minor themes were also explored.” Such as? Are the authors talking about the micro experiences in line 230?

Reviewer #5: Objectives of the study are identified with a hypothesis being tested.

Samples size: sufficient for this study.

Statistical analysis support conclusions.

Major revision:

- The first part of the introduction should be revised, see below.

- The authors should mention how study participation was approved in terms of illiteracies, e.g. thumb print. No further concerns about ethical clearance in general.

Minor revision:

- “WHO” throughout the document should be spelled as “the WHO”.

**Results**

-Does the analysis presented match the analysis plan?

-Are the results clearly and completely presented?

-Are the figures (Tables, Images) of sufficient quality for clarity?

Reviewer #4: - The analysis presented matches the overall analysis plan.

- Concerning the presentation of the results and the quality of figures, I suggest the authors to add a summary table of the findings, with one column for BU and another for yaws, including shorter examples of the messages conveyed by participants in each of the dimensions assessed and the percentages of patients that were included in them. The manuscript is lenghty and it would help to better convey the message.

Point-by-point:

- Lines 238-240: In other studies in BU in African countries, category I/II lesions are the most frequently found. Why are there so many category III lesions in your population when compared to other studies? What is the impact that this has on healthcare seeking and stigma?

- Lines 265-270 and 551-560: see the previous point.

- Lines 426-427: “For participants with yaws, there was no observed effect of stigma on social relationships.” Yet, the authors never discuss hypotheses for this result nor why it could be different from BU.

- Lines 540-543 need revision:

 o “Further, stigma experiences differ among children and adults.” In BU? In yaws? In both?

 o “Stigma experiences contributed largely to negative outcomes.” Since this is a qualitative study, either this sentence should rephrase to “were perceived in this study to contribute…” or otherwise citations from previous works are needed to back it.

 o “Participants who used religious coping reported higher psychosocial suffering as compared to those who coped using active mechanisms such as confrontation.” There was no formal statistical analysis of this, meaning that this shouldn’t be written so bluntly. An alternative would be “tended to report”.

- Lines 582-589: it is the only time in the discussion that Goffman theory was mentioned, although it guided the way the work was approached. Shouldn’t this be more discussed?

- Lines 594-596: authors state differences between their observations and previous results regarding the role of religion in managing stigma, but no discussion is provided on the why behind those differences. If comparisons are being made, they should be discussed, even if shortly.

Reviewer #5: Analysis presented per plan. Results are clearly presented, and tables are in order.

**Conclusions**

-Are the conclusions supported by the data presented?

-Are the limitations of analysis clearly described?

-Do the authors discuss how these data can be helpful to advance our understanding of the topic under study?

-Is public health relevance addressed?

Reviewer #4: - The conclusions summarize the data presented.

- The limitations of analysis are described.

- The way authors discuss how the data can be helpful to advance our understanding of the topic under study does not always match the data presented (please refer to the point-by-point discussion). Furthermore, the omission of relevant references from the literature in the introduction and discussion make it difficult to ascertain the novelty of the findings, which appears to be limited as it is currently presented.

- Public health relevance is addressed, albeit it could be expanded with more concrete actions that could be taken to address the various issues identified. Additionaly, as the authors mention, there were studies on stigma already performed in Ghana. Were there programs estabilished to address stigma after those studies were published? If yes, could the data the authors now present influence those programs in any way?

Point-by-point:

- Lines 646-648: in what way would the authors use the newly acquired knowledge to design and integrate strategies to address stigma?

- Lines 651-653: authors mention the role of teachers and their role in addressing the negative impact of stigma. Yet, this was never discussed during the manuscript and the few times that teachers were mentioned in the results section, they were actually part of the problem and not of the solution. So, what is the role of teachers?

Reviewer #5: see summary and comments

**Editorial and Data Presentation Modifications?**

Reviewer #4: - Lines 201-203: This line should have been in the previous paragraph.

- Line 224: extra blank space in “to ensure”

Reviewer #5: none, besides above mentioned

**Summary and General Comments**

Reviewer #4: I thank the authors for their manuscript “Stigma experiences, effects and coping among individuals affected by Buruli ulcer and yaws in Ghana”, which focuses on the stigma perceived by individuals afflicted by Buruli ulcer (BU) or yaws. This is a pertinent topic, as previous studies indicated it may by a source of additional mental distress, with potential impact on disease outcomes. To this aim, authors refer to the Goffman’s theory on stigma to conduct a series of interviews with BU and yaws patients, thereby enabling them to identify disparities in the stigma perceived by patients in each of these conditions.

Even though this is the main message of the study and worthy of reflection, there is not a clear alignment between ideas, from the problems raised in the introduction, to the overall objective of the study and the discussion of the results. Furthermore, the manuscript would benefit from a sharper definition of the primary strengths of the approach taken and from a more extended revision of its discussion section, as the potential impact of the findings with more concrete examples is not sufficiently addressed.

I commend the authors for the effort they made in collecting their data and hope to see their manuscript at its best, if not in this journal, in another.

Reviewer #5: The paper is – in general - well written, and analyses seem adequate. 

Nevertheless, I would suggest rephrasing the first paragraph of the introduction as this seems to me not adequate for the general reader. It is written too specific and does not allow an insight of the analysed diseases for the reader who is not practiced in these NTD’s. 

Stigma related to NTDs were not yet published on a broad range – publications were limited to a small number of NTDs. Stigmata constitute a large burden on social and economic life of affected patients and other involved persons/communities. 

The authors are going to publish a very important article as recommended by the WHO.

PLOS authors have the option to publish the peer review history of their article (what does this mean?). If published, this will include your full peer review and any attached files.

Reviewer #4: No

Reviewer #5: Yes: PD Dr. Marcus Beissner
---

## [Editor Report · Decision Letter 2]

22 Mar 2024

Dear Amoako,

We are pleased to inform you that your manuscript 'Stigma experiences, effects and coping among individuals affected by Buruli ulcer and yaws in Ghana' has been provisionally accepted for publication in PLOS Neglected Tropical Diseases.

Best regards,

Elsio A Wunder Jr, DVM, Ph.D.

Section Editor

---

## [Editor Report · Acceptance letter]

22 Apr 2024

Dear Amoako,

We are delighted to inform you that your manuscript, "Stigma experiences, effects and coping among individuals affected by Buruli ulcer and yaws in Ghana," has been formally accepted for publication in PLOS Neglected Tropical Diseases.

Best regards,

Shaden Kamhawi

co-Editor-in-Chief

Paul Brindley

co-Editor-in-Chief
